

# Genetic structure and historical and contemporary gene flow of *Astyanax mexicanus* in the Gulf of Mexico slope: a microsatellite-based analysis

Rodolfo Pérez-Rodríguez[1], Sarai Esquivel-Bobadilla[2], Adonaji Madeleine Orozco-Ruíz[2], José Luis Olivas-Hernández[2] and Francisco Javier García-De León[2]

[1] Laboratorio de Biología Acuática, Facultad de Biología, Universidad Michoacana de San Nicolás de Hidalgo, Morelia, Michoacán, México

[2] Laboratorio de Genética para la Conservación, Centro de Investigaciones Biológicas del Noroeste, La Paz, Baja California Sur, México

## ABSTRACT

**Background**. *Astyanax mexicanus* from the river basins of the Gulf of Mexico slope are small freshwater fish that usually live in large groups in different freshwater environments. The group is considered successful due to its high capacity for dispersal and adaptation to different habitats, and the species present high morphological variability throughout their distribution in Mexico. This has produced the most extreme morphotype of the group; the hypogeous or troglobite, which has no eyes or coloration, and is probably the cause of taxonomic uncertainty in the recognition of species across the entire range. Most studies of *A. mexicanus* have mainly focused on cave individuals, as well as their adjacent surface locations, providing an incomplete evolutionary history, particularly in terms of factors related to dispersal and the potential corridors used, barriers to gene flow, and distribution of genetic variability. The aim of the present study is to determine the population structure and the degree and direction of genetic flow in this complex taxonomic group, incorporating geographic locations not previously included in analyses using microsatellite loci. Our aim is to contribute to the knowledge of the intricate evolutionary history of *A. mexicanus* throughout most of its range.

**Methods**. The present study included a set of several cave and surface locations of *A. mexicanus*, which have been widely sampled along the Gulf of Mexico slope, in a genetic population analysis using 10 microsatellite loci.

**Results**. Ten genetic populations or lineages were identified. In these populations, gene flow was recorded at two time periods. Historical gene flow, both inter and intra-basin, was observed among surface populations, from surface to cave populations, and among cave populations, whereas recording of contemporary gene flow was limited to intra-basin exchanges and observed among surface populations, surface to cave populations, and cave populations.

Corresponding author
Francisco Javier García-De León, fgarciadl@cibnor.mx

## INTRODUCTION

The Mexican characid *Astyanax* is a Neotropical genus found in North American basins, reaching as far north as the Bravo River hydrographic system (*Strecker, Faundez & Wilkens, 2004*). This group includes several species for which the taxonomy is uncertain and currently in dispute, even though various morphological and molecular genetic approaches have been applied to the taxon (*Schmitter-Soto, 2017*; *Wilkens & Strecker, 2017*). *Astyanax* from the river basins of the Gulf of Mexico slope refer to a group of three lineages that are mitochondrially well-differentiated and correspond to *Astyanax mexicanus*, *Astyanax aeneus*, and *Astyanax hubbsi* (*Ornelas-García, Domínguez-Domínguez & Doadrio, 2008*; *Ornelas-García & Pedraza-Lara, 2016*). The first two taxa coincide with a geographic congruence along the Gulf of Mexico slope, reflecting a vicariant pattern (*Miller, 1986*), i.e., the taxon located north of the Trans Mexican Volcanic Belt (TMVB) corresponds to *A. mexicanus*, while the taxon located south of the TMVB corresponds to *A. aeneus*, although both species co-occur in a contact zone along the boundaries of their distribution ranges (*Ornelas-García, Dominguez-Dominguez & Doadrio, 2008*; *Ornelas-García & Pedraza-Lara, 2016*). In the case of *A. hubbsi*, *Ornelas-García & Pedraza-Lara (2016)* consider it as a relict of the older expansion of *Astyanax* in North America. However, hybridization detected with microsatellite markers has been reported between *A. mexicanus* and *A. aeneus*, and between *A. mexicanus* and *A. hubbsi*, calling into question the validity of the species as independent evolutionary units (*Hausdorf, Wilkens & Strecker, 2011*). The recurrent introgression recorded among these three species or lineages of *Astyanax* of the Gulf of Mexico slope is regarded as the main factor that renders species delimitation based on a single marker (*Ornelas-García, Dominguez-Dominguez & Doadrio, 2008*; *Ornelas-García & Pedraza-Lara, 2016*) a difficult task (*Hausdorf, Wilkens & Strecker, 2011*; *Herman et al., 2018*). With a more cautious and operational approach, *Torres-Paz et al. (2018)* assign all surface and cave lineages of *Astyanax* spp. of the Gulf of Mexico slope to *A. mexicanus*; therefore, in this study this approach was applied.

In addition to the reticulate genetic pattern, *A. mexicanus* shows a high morphological plasticity strongly associated with recurrent morphological convergence (parallel evolution) to similar environments (*Ornelas-García, Dominguez-Dominguez & Doadrio, 2008*; *Garita-Alvarado, Barluenga & Ornelas-García, 2018*). The most studied and conspicuous morphotype in *A. mexicanus* is the cave morph (blind and depigmented, *Gross, 2012*; *Garita-Alvarado, Barluenga & Ornelas-García, 2018*). Studies on population genetics in *A. mexicanus* have produced several important findings, most of which focus on cavefish populations (*Gross, 2012*). These studies show that the cavefish populations are derived mainly from several distinct temporary cave invasions (*Dowling, Martasian & Jeffery, 2002*; *Strecker, Bernatchez & Wilkens, 2003*; *Ornelas-García & Pedraza-Lara, 2016*). These invasions have left signs of different degrees of troglomorphy in the cave populations, from individuals with pigmentation and some functional, but slightly reduced, visual systems, to those with pigmentation or visual systems that are significantly reduced or absent (*Strecker, Faundez & Wilkens, 2004*; *Bradic et al., 2012*). As a result of these events, some cave populations remain completely isolated, while others indicate gene flow with

the surface locations (*Bradic et al., 2012*). Nevertheless, it should be noted that most of these studies feature partial sampling, limited focus and/or different markers. Other sets of morphological traits that reveal parallel evolution are the body shape and the trophic traits of the head that have been associated with the trophic specialization (tooth shape, dental formula, eye size, snout length, body depth, head profile and mouth orientation) of two species from two independent lakes (*Garita-Alvarado, Barluenga & Ornelas-García, 2018*; *Ornelas-García et al., in press*). But unlike cave morphs, there are few population genetic studies that limit themselves to assessing the genetic differentiation between two lacustrine divergent morphs isolated in sympatry within Catemaco Lake, which were originally considered to be different genera (i.e., *Bramocharax* and *Astyanax*), and currently, are even considered to be different species (*A. aeneus* and *Astyanax caballeroi*) (*Ornelas-García, Bastir & Doadrio, 2014*). Therefore, it remains unknown whether the population has historical gene flow with specimens from other nearby basins.

*Astyanax mexicanus* constitute an excellent study model group, not only because of their widely studied traits associated with parallel evolution, but also due to their high capacity for dispersal across a wide geographic area that has allowed colonization of different habitats and the occurrence of distinct recurrent introgression events over time (*Hausdorf, Wilkens & Strecker, 2011*; *Strecker, Hausdorf & Wilkens, 2012*; *Bradic et al., 2012*; *Coghill et al., 2014*). Determination of the magnitude and direction of both the historical and contemporary gene flow of one of the freshwater fish species distributed along the Gulf of Mexico slope therefore represents an important contribution to our knowledge of one of the transition zones of aquatic fauna between Neotropical and Nearctic regions (*Miller, 1986*; *Obregón-Barboza, Contreras-Balderas & Lozano-Vilano, 1994*). In addition to the land barriers situated among the different river basins that lie along the Gulf of Mexico slope, there is a marked geographic barrier consisting of a volcanic mountain range. This mountain range forms the eastern limit of the TMVB known as Punta del Morro (PDM), which since its formation has served as a major geographic barrier for several freshwater fish groups (*Obregón-Barboza, Contreras-Balderas & Lozano-Vilano, 1994*; *Contreras, Obregón & Lozano, 1996*; *Mateos, Sanjur & Vrijenhoek, 2002*; *Perdices et al., 2002*; *Hulsey et al., 2004 Ornelas-García, Dominguez-Dominguez & Doadrio, 2008*; *Agorreta et al., 2013*).

The aim of this study is to integrate genetic information from new combination set of microsatellite markers using surface and cave populations that have never been analyzed together, in order to contribute to the evolutionary history of this characid fish. We wish to know how many genetic populations or lineages exist in the different hydrographic basins of the Mexican Atlantic slope. Furthermore, we wish to know whether the hydrological basins of the Mexican Atlantic slope and the TMVB, represent barriers against gene flow for *Astyanax mexicanus*, a highly dispersive species. The PDM is expected to function as a barrier to genetic flow despite the high dispersive capacity of the species and the direction of gene flow is greater in surface locations close to the caves than those that are further away. By including samples taken north of the location of the cave populations, we seek to determine whether another, previously undetected, colonization event has taken place in the caves.

**Table 1  Sampling localities of _A. mexicanus_ and their respective basins.**

| Location | Acronym | Mitochondrial assignment | n | Latitude | Longitude | Basin |
|---|---|---|---|---|---|---|
| Cuatro Ciénegas | CC | _A. mexicanus_ | 26 | 26.878 | −102.137 | Bravo |
| San Fernando | SF | _A. mexicanus_ | 30 | 24.431 | −98.431 | San Fernando |
| Garza Valdez | GV | _A. mexicanus_ | 24 | 24.711 | −99.384 | Soto la Marina |
| Troncones | TR | _A. mexicanus_ | 40 | 23.723 | −99.308 | Soto la Marina |
| Arroyo Lagartos | AL | _A. mexicanus_ | 36 | 22.804 | −98.941 | Pánuco |
| Molino Cave | ML | _A. mexicanus_ | 15 | 23.031 | −99.150 | Pánuco |
| Pachón Cave | PCH | _A. mexicanus_ | 44 | 22.607 | −99.044 | Pánuco |
| Sabinos Cave | SAB | _A. hubbsi_ | 23 | 22.024 | −98.9 | Pánuco |
| Tinajas Cave | TIN | _A. hubbsi_ | 12 | 21.968 | −98.902 | Pánuco |
| La Cañada | LCA | _A. mexicanus_ | 26 | 21.867 | −99.151 | Pánuco |
| Huichihuayan | SLP | _A. mexicanus_ | 19 | 21.479 | −98.966 | Pánuco |
| El Zapotal | V | _A. aeneus_ | 30 | 21.276 | −98.146 | Túxpan |
| Catemaco | CAT | _A. aeneus_[a] | 26 | 18.433 | −95.032 | Papaloapan |
| Teapa | TE | _A. aeneus_ | 50 | 17.426 | −92.751 | Grijalva-Usumacinta |
| Tapijulapa | TA | _A. aeneus_ | 27 | 17.45 | −92.75 | Grijalva-Usumacinta |
| Rio Tzendales | RTZ | _A. aeneus_ | 41 | 16.216 | −90.841 | Grijalva-Usumacinta |

Notes.

   _n_, sample size. Mitochondrial assignment refers to the presumed species to which the literature assigns those sampling sites, see text for more explanation.

   [a]Previously considered _Bramocharax caballeroi_.

## MATERIALS & METHODS

### Sampling

The present study included samples of _Astyanax_ spp. from at least seven of the main hydrographic basins of the Gulf of Mexico slope in Mexico: The Bravo, San Fernando, Soto la Marina, Pánuco, Tuxpan, Papaloapan, and Grijalva-Usumacinta River basins (Table 1; Fig. 1). Fish specimens were collected using electrofishing equipment and trawl nets. Tissue samples (fin clips) were preserved in 96% ethanol for subsequent DNA extraction. A total of 469 individuals from 16 localities, corresponding to 4 cave and 12 surface locations, were included. Collection of the samples was carried out with the approval of the Mexican government (DGOPA.05003.181010-5003; DGOPA.00570.288108-0291; DAPA/2/130409/0961, 230401-613-03).

### DNA extraction, PCR and genotyping

Total DNA was extracted from fin tissue using a DNeasy blood and tissue kit from Qiagen. Ten polymorphic microsatellite loci developed for the genus _Astyanax_ (_Strecker, 2003_; _Protas et al., 2006_) were studied (Table S1) through amplification in 15 μl reactions, following the cycling conditions shown in Table S2. Individual genotyping was obtained using 6% denaturing polyacrylamide electrophoresis gels, run in a vertical electrophoresis unit at 1,700 V, 40A and 45 W for 2.5 h. DNA fragments were visualized by silver staining (_Bassam & Caetano-Anollés, 1993_). Allelic size was obtained with gels using the Sequentix Gel Analyzer (http://www.sequentix.de/).

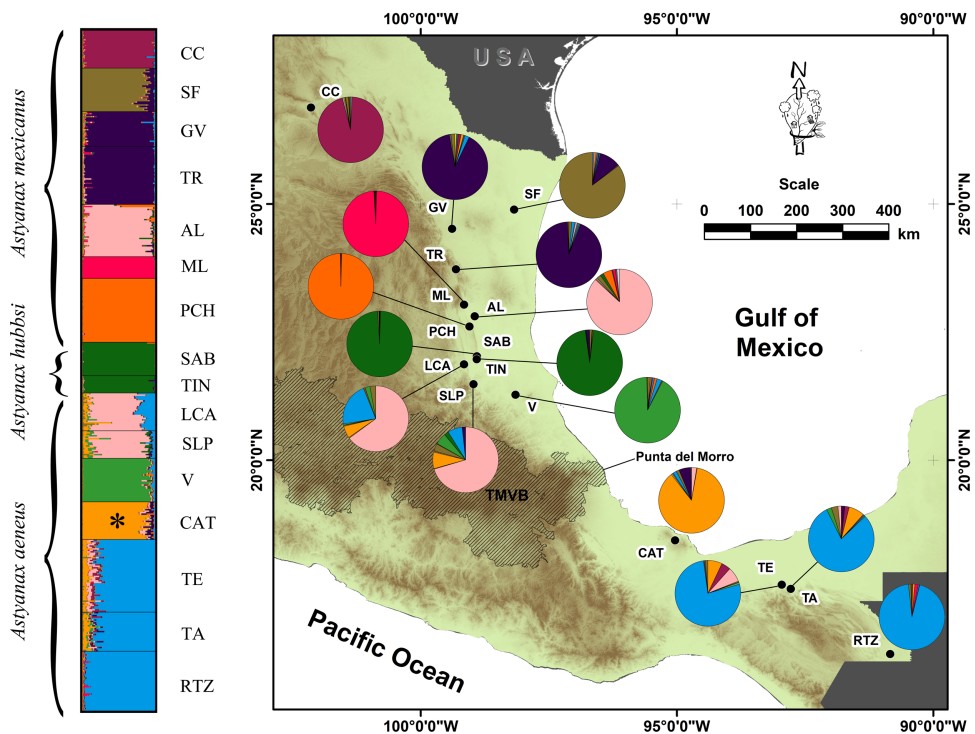

**Figure 1** **Geographic distribution of the genetic variation in populations and locations of *A. mexicanus* sampling in this study along the Gulf of Mexico slope.** Clustering analysis performed in STRUCTURE, using 10 microsatellite loci of *A. mexicanus* sampled; the phylogenetic classification *sensu Ornelas-García, Dominguez-Dominguez & Doadrio (2008)* and resulting populations were matched. The map shows the distribution range sampled in the present study; colored pies charting on the map corresponds to the same genotypic clusters of STRUCTURE. For definition of the acronyms see Table 1.

## Statistical analyses

In order to detect the presence of null alleles at every locus and in each location, the Brookfield 1 Index (*Brookfield, 1996*) and the Expectation-Maximization algorithm (*Dempster, Laird & Rubin, 1977*) were applied using MICRO-CHECKER version 2.2.3 (*Oosterhout, Wills & Hutchinson, 2004*) and FreeNA (*Chapuis & Estoup, 2007*), respectively. Linkage disequilibrium and significant deviation from Hardy-Weinberg (HW) expectations using an exact test (10,000 dememorization steps, 5,000 batches of 10,000 iterations per batch) were determined with GENPOP (*Raymond & Rousset, 2001*). Statistical significance values were corrected using the sequential Bonferroni method (*Rice, 1989*), with a significance value of 5%.

Genetic variability within populations (as estimated by STRUCTURE) was evaluated by the number of alleles (N), number of alleles per locus (Na), effective number of alleles (Nae), observed heterozygosity ($H_O$), and the fixation index ($F_{IS}$), using GENETIX 4.05 (*Belkhir et al., 1999-2004*). Unbiased expected (He) heterozygosity was calculated in GenAlex 6.5 (*Peakall & Smouse, 2012*). Allelic richness ($A_R$) was calculated in the function allele.rich in PopGeReport. v3.0.4 (*Adamack & Gruber, 2014*), in R (*R Core Development Team, 2017*), which uses the rarefaction approach to deal with differences in sample sizes.

## Clustering and genetic differentiation

The fixation index $F_{ST}$, based on *Weir & Cockerham (1984)*, was estimated with GENETIX 4.05 (*Belkhir et al., 1999-2004*) to describe genetic differentiation based on the variance in allele frequencies among locations. Bonferroni corrections (adjusted $P = 0.05$) were performed by multiple assessment (*Rice, 1989*). In addition, $F_{ST}$ were estimated using and not using ENA algorithm as implemented in FreeNA.

A Discriminant Analysis of Principal Components (DAPC) was performed using the ADEGENET package (*Jombart, 2008*) in *R Core Development Team, (2017)*, through cross-validation of the POPPR package (*Jombart, Devillard & Balloux, 2010*) with the xvalDapc function. This function chooses the number of axes to retain by testing different numbers of PCs, and the DAPC quality is subsequently evaluated by cross-validation. The DAPC is carried out on the set comprised of 90% of the observations, and then used to predict the groups of the remaining 10% of observations. The number of PCs associated with the lowest Mean Squared Error is then retained for the DAPC. This method combines the advantages of principal component analysis, by assuming neither Hardy-Weinberg equilibrium nor linkage disequilibrium (*Jombart, 2008*), and from discriminant analysis, by which it attempts to summarize the genetic differentiation between groups, while omitting variation within groups (*Jombart, Devillard & Balloux, 2010*). Two independent analyses were carried out, one using all 16 locations and the other using only the 12 surface locations, while excluding the four cave locations.

Analysis of molecular variance (AMOVA) was conducted for four different partitions of the 16 locations using ARLEQUIN 3.5 (*Excoffier & Lischer, 2010*). The following four grouping models were employed under different criteria to maximize variance between the groups and thus reveal the genetic structure contained in the data: (A) All 16 locations; (B) phylogenetic criteria considering two subclades of *A. mexicanus* [(CC)+(SF, GV, TR, AL, ML, PCH, LCA, SLP, V)], subclades of *A. aeneus* [(CAT), (TE, TA, RTZ)], and the clade of *A. hubbsi* (SAB, TIN); (C) hydrographic basin criteria considering the seven independent river basins included in the present study [Bravo (CC), San Fernando (SF), Soto la Marina (GV, TR), Pánuco (AL, ML, PCH, SAB, TIN, LCA, SLP), Túxpan (V), Papaloapan (CAT), Grijalva-Usumacinta (TE, TA, RTZ)]; (D) Localities were grouped according to their degree of genetic differentiation as measured by the $F_{ST}$, which resulted in the grouping of those locations that presented non-significant differentiation levels [(CC), (SF), (GV, TR), (AL, LCA, SLP), (V), (PCH), (ML), (SAB, TIN), (CAT), (TE, TA, RTZ)]. Ten thousand permutations were used in each of these analyses.

A Bayesian model-based clustering method was also performed using STRUCTURE 2.3.3 software (*Pritchard, Stephens & Donnelly, 2000*; *Pitchard, 2007*) in order to evaluate population structure and assign individuals to genetic clusters. This approach was conducted using the admixture model, which excluded information on the location of origin and assumed independence among loci and non-informative priors. Values were tested for $K = 1$ to $K = 16$, including all locations, and for $K = 1$ to $K = 12$, without cave locations, that is, including only surface locations. The mean and variance of the log likelihoods for each $K$ were calculated from 10 independent runs of 100,000 iterations in order to determine the highest posterior probability $K$. To estimate the true number of

clusters represented along the sampled range of *Astyanax* spp. (*Evanno, Regnaut & Goudet, 2005*), $\Delta K$ statistics were calculated by http://taylor0.biology.ucla.edu/struct_harvest/ (*Earl, 2011*). Individual and mean population membership coefficients of ancestry in our inferred demes were presented graphically in DISTRUCT, version 1.1 (*Rosenberg, 2004*).

### Isolation by distance and historical contemporary gene flow

The pattern of isolation-by-distance (IBD) among locations was assessed by performing a Mantel test, based on pairwise $F_{ST}$ and geographic distances between distinct locations, using the Isolde algorithm with the on-line version of GENEPOP (http://genepop.curtin.edu.au/). Four distinct Mantel tests were performed as follows: (1) using all 16 locations; (2) using only the surface populations; (3) using only the locations located north of the TMVB and (4) using only the locations south of the TMVB. A total of 10,000 permutations were required to estimate the 95% upper tail probability of the matrix correlation coefficients.

Historical gene flow (for approximately the four previous $N_e$ generations (*Beerli & Felsenstein, 2001*) was estimated using the program MIGRATE-N 3.6.11 (*Beerli & Felsenstein, 2001*). Mutation-scaled migration rates (M = m/μ, where m = migration rate and μ = mutation rate) and mutation-scaled size theta ($\Phi = 4N_e\mu$, where $N_e$ = effective population size, and μ = mutation rate) were estimated between locations. Different migration model parameter distributions were used between every location detected by MIGRATE. Microsatellite mutation was modeled as a continuous Brownian process and the mutation rate was set to constant for all loci. A static heating scheme was used with four chains and temperatures (1.0, 1.5, 3.0 and 1000000), for which the initial $\Phi$ and m values were generated from the $F_{ST}$ option. Exponential priors were placed for both $\Phi$ (bounded between 0 and 50.0) and M (bounded between 0 and 100.0). The analyses were conducted using 10 replicates of a single long Markov chain, with 100,000 steps recorded for every 100 generations and the first 500,000 trees per run discarded as burn-in. Three patterns of migration were estimated in *A. mexicanus* from the Gulf of Mexico slope: among surface populations, among cave populations, and from surface to cave. Gene flow from cave to surface has been regarded as a nonexistent event, as reflected in its negligible estimate ($1.5 \times 10^{-6}$ to $10.2 \times 10^{-4}$, *Fumey et al., 2018*).

Recent gene flow (in the past few generations) was estimated using BAYESASS 3.04 (*Wilson & Rannala, 2003*). This program estimates the posterior probability of an individual's migratory history and thus allows estimation of the rate and direction of recent dispersal. The Markov Chain Monte Carlo method was run for 10,000,000 iterations. Delta values (i.e., maximum parameter change per iteration) were left as default (*Beerli & Felsenstein, 2001*).

## RESULTS

Hardy-Weinberg equilibrium tests following Bonferroni correction revealed heterozygote excess at the following loci and locations: locus *Am145a* for GV/TR, locus *Ast02* for AL/LCA/SLP and SAB/TIN, and loci *Ast10* and *Am106b* for CAT. Nevertheless, the HW multilocus did not show deviation from equilibrium (Table S3). MICRO-CHECKER

**Table 2 Pairwise $F_{ST}$ between *Astyanax* locations.**

|      | CC   | SF   | GV    | TR    | AL   | ML   | PCH  | SAB  | TIN  | LCA  | SLP  | V    | CAT   | TE    | TA    | RTZ  |
|------|------|------|-------|-------|------|------|------|------|------|------|------|------|-------|-------|-------|------|
| CC   |      | 0.27 | 0.25  | 0.25  | 0.21 | 0.59 | 0.66 | 0.53 | 0.49 | 0.23 | 0.23 | 0.29 | 0.24  | 0.16  | 0.18  | 0.18 |
| SF   | 0.27 |      | 0.19  | 0.18  | 0.21 | 0.56 | 0.63 | 0.49 | 0.42 | 0.20 | 0.19 | 0.24 | 0.20  | 0.16  | 0.17  | 0.18 |
| GV   | 0.26 | 0.20 |       | 0.01  | 0.14 | 0.45 | 0.57 | 0.44 | 0.38 | 0.17 | 0.12 | 0.18 | 0.09  | 0.13  | 0.12  | 0.15 |
| TR   | 0.26 | 0.18 | 0.01* |       | 0.14 | 0.42 | 0.52 | 0.40 | 0.35 | 0.15 | 0.12 | 0.17 | 0.10  | 0.13  | 0.13  | 0.15 |
| AL   | 0.21 | 0.21 | 0.14  | 0.13  |      | 0.43 | 0.45 | 0.33 | 0.27 | 0.05 | 0.05 | 0.10 | 0.12  | 0.07  | 0.08  | 0.09 |
| ML   | **0.58** | **0.57** | **0.44** | **0.41** | **0.4** |      | 0.85 | 0.81 | 0.81 | 0.47 | 0.49 | 0.50 | 0.43  | 0.40  | 0.45  | 0.43 |
| PCH  | **0.67** | **0.64** | **0.57** | **0.52** | **0.45** | *0.85* |      | 0.75 | 0.76 | 0.50 | 0.54 | 0.52 | 0.53  | 0.44  | 0.52  | 0.48 |
| SAB  | **0.55** | **0.51** | **0.45** | **0.41** | **0.34** | *0.82* | *0.77* |      | 0.27 | 0.38 | 0.37 | 0.38 | 0.39  | 0.34  | 0.37  | 0.36 |
| TIN  | **0.50** | **0.44** | **0.39** | **0.36** | **0.27** | *0.8* | *0.77* | 0.27 |      | 0.30 | 0.29 | 0.30 | 0.33  | 0.29  | 0.31  | 0.30 |
| LCA  | 0.24 | 0.20 | 0.17  | 0.15  | 0.05 | **0.47** | **0.51** | **0.4** | **0.31** |      | 0.07 | 0.11 | 0.11  | 0.06  | 0.07  | 0.06 |
| SLP  | 0.23 | 0.19 | 0.12  | 0.12  | 0.05 | **0.48** | **0.54** | **0.39** | **0.3** | 0.07 |      | 0.09 | 0.10  | 0.06  | 0.07  | 0.07 |
| V    | 0.30 | 0.25 | 0.19  | 0.18  | 0.11 | **0.49** | **0.53** | **0.4** | **0.31** | 0.12 | 0.1  |      | 0.14  | 0.11  | 0.11  | 0.12 |
| CAT  | 0.24 | 0.20 | 0.09  | 0.1   | 0.12 | **0.42** | **0.53** | **0.4** | **0.33** | 0.12 | 0.1  | 0.14 |       | 0.07  | 0.07  | 0.08 |
| TE   | 0.16 | 0.16 | 0.13  | 0.13  | 0.07 | **0.39** | **0.44** | **0.35** | **0.3** | 0.05 | 0.06 | 0.11 | 0.07  |       | 0.01  | 0.01 |
| TA   | 0.18 | 0.17 | 0.13  | 0.14  | 0.08 | **0.45** | **0.52** | **0.38** | **0.32** | 0.08 | 0.07 | 0.11 | 0.07  | 0.01* |       | 0.02 |
| RTZ  | 0.19 | 0.18 | 0.15  | 0.15  | 0.09 | **0.43** | **0.49** | **0.37** | **0.31** | 0.06 | 0.08 | 0.13 | 0.08  | 0.01* | 0.02* |      |

**Notes.**
*Non-significant values.
Values in italic font are the highest; values in bold only are the second highest; black normal font values are moderate and gray normal font values are low but significant. For definition of the acronyms, see Table 1. Diagonal upper values correspond to $F_{ST}$ values estimated by ENA algorithm as implemented in FreeNA.

detected null alleles in only two tests (locus *Ast10* for CAT, and locus *Ast02* for SAB-TIN), and FreeNA detected the presence of null alleles at five loci in different genetic populations: locus *Ast09* for CC, locus *Am241b* for SF, locus *Am145a* for GV/TR, loci *Am122b* and *Am106b* for AL/LCA/SLP, and locus *Am106b* for CAT, but, none of them at frequencies greater than 0.15 (Table S4). There was no evidence of linkage disequilibrium among pairs of loci across all populations (Table S5).

## Population structure

Pairwise $F_{ST}$ values revealed high, moderate, and low divergences (Table 2). High and significant values were observed among the cave locations (0.77–0.85), except in the comparison between the SAB and TIN caves (0.27). Moderate and significant values were recorded for most of the pairwise comparisons of surface locations, regardless of geographic distance between them (0.11–0.30), while low but significant values were found for the comparisons of surface locations of the Pánuco basin (AL, LCA, SLP) with those of the Papaloapan (CAT) and Grijalva-Usumacinta (TE, TA, RTZ) basins (0.05–0.10). The comparisons made within the Soto la Marina (GV vs TR) and Grijalva-Usumacinta (TE vs TA vs RTZ) basins presented the lowest and non-significant values (0.01–0.02). Pairwise $F_{ST}$ values using and not using ENA did not present significant differences (Table S6).

An initial DAPC, with all 16 locations, showed the formation of four main clusters (Figs. 2A and 2B) in which 15 discriminant functions were retained. The first two functions accounted for 93.3% of the total variance. Cluster *a* corresponded to the PCH cave; cluster *b* included the SAB and TIN caves; cluster *c* was formed by the ML cave, and
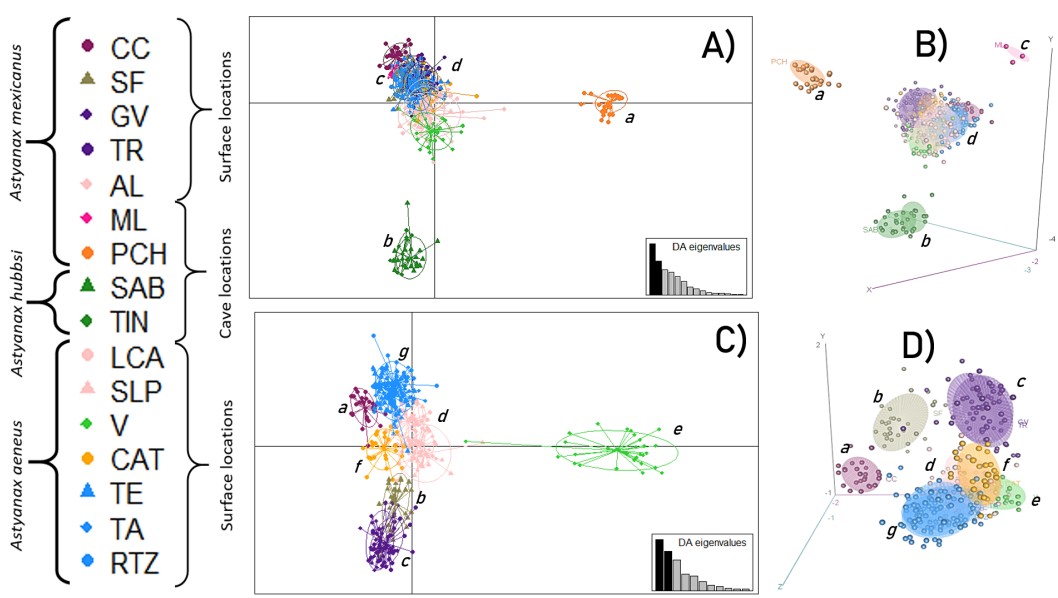

**Figure 2 Discriminant Analysis of Principal Components (DAPC).** (A and B) Analysis including all 16 locations; (C and D) Analysis of 12 populations (excluding the cave populations). The phylogenetic classification *sensu Ornelas-García, Dominguez-Dominguez & Doadrio (2008)*. For definition of the acronyms see Table 1.

cluster *d* corresponded to all the surface locations. A second DAPC, with only the 12 surface locations, revealed the formation of seven clusters (Figs. 2C and 2D) in which 11 discriminant functions were retained. The first two functions accounted for 94% of the total variance. Cluster *e* was formed by the CC location; cluster *f* by the SF location; cluster *g* included the GV and TR locations; cluster *h* was formed by the AL, LCA, and SLP locations; cluster *i* included the V location; cluster *j* included the CAT location; and finally, cluster *k* included the remaining surface locations, TE, TA, and RTZ.

The AMOVA analysis indicated that model D of 10 populations showed the highest intergroup variance ($F_{CT} = 0.2449$, $P = 0.00$) and minimal intra-group variance ($F_{SC} = 0.0455$, $P = 0.00$) (Table 3). STRUCTURE also revealed 10 genetic groups. Three cave populations were assigned the greatest probability of ancestry (ML: 99%, PCH: 99%, SAB-TIN: 97%), as were the following epigean populations corresponding to distinct river basins: Bravo (CC: 95%), Soto la Marina (GV+TR: 92%), and Tuxpan (V: 91%). Other populations that also were homogeneous, but with lower assignment probability, were the Papaloapan (CAT: 85%), Grijalva-Usumacinta (TE+TA+RTZ: 84%), and Pánuco (AL+LCA+SLP: 70%) basins (Fig. 1). The Pánuco and Grijalva-Usumacinta basins presented a genetic admixture between them (Fig. 1), which was reflected in both analyses (DAPC and $F_{ST}$ values) (Fig. 2; Table 2). Another genetic admixture was recorded between the populations of the Soto la Marina and Papaloapan basins (Fig. 1).

In summary, the results of $F_{ST}$, DAPC, AMOVA, and STRUCTURE agree that the number of homogeneous genetic groups along the Atlantic slope is ten, consisting of three cave populations, and seven surface populations.

**Table 3  Four distinct AMOVA models assessed.**

| Tested model | Source of variation | Variance | Percentage of the total (%) | Statistics F | P value |
|---|---|---|---|---|---|
| A). 16 groups | Between groups | 1.16376 | 26.55 | | |
| | | | | $F_{ST} = 0.2654$ | **0.00** |
| | Within populations | 3.21974 | 73.45 | | |
| B). 5 groups | Between groups | 0.29026 | 6.47 | $F_{CT} = 0.0647$ | 0.13 |
| | Within groups | 0.97395 | 21.72 | $F_{SC} = 0.2322$ | **0.00** |
| | Within populations | 3.21974 | 71.81 | $F_{ST} = 0.2819$ | **0.00** |
| C). 7 groups | Between groups | 0.12916 | 2.93 | $F_{CT} = 0.0293$ | 0.30 |
| | Within groups | 1.05758 | 24.00 | $F_{SC} = 0.2472$ | **0.00** |
| | Within populations | 3.21974 | 73.07 | $F_{ST} = 0.2693$ | **0.00** |
| D). 10 groups | Between groups | 1.09414 | 24.49 | $F_{CT} = 0.2449$ | **0.00** |
| | Within groups | 0.15367 | 3.44 | $F_{SC} = 0.0455$ | **0.00** |
| | Within populations | 3.21974 | 72.02 | $F_{ST} = 0.2793$ | **0.00** |

**Notes.**

Median values (with their 0.025 and 0.975 posterior distribution values as 95% confidence interval estimates) of M = migration rate and Φ = size theta. Values in bold correspond to higher M values. For definition of the acronyms, see Table 1.

(A) Each collection location considered as a different population (CC, SF, GV, TR, AL, ML, PCH, SAB, TIN, LCA, SLP, V, CAT, TE, TA and RTZ).

(B) Localities were grouped by phylogenetic groups based on the results obtained from *Ornelas-García, Dominguez-Dominguez & Doadrio (2008)* [(CC), (SF, GV, TR, AL, ML, PCH, LCA, SLP, V), (SAB, TIN), (CAT), (TE, TA, RTZ)].

(C) Localities were grouped by hydrological basins [Bravo (CC), San Fernando (SF), Soto la Marina (GV, TR), Pánuco (AL, ML, PCH, SAB, TIN, LCA, SLP), Tuxpan (V), Papaloapan (CAT), Grijalva-Usumacinta (TE, TA, RTZ)].

(D) Localities were grouped according to their degree of genetic differentiation as measured by the $F_{ST}$, grouping those populations that presented non-significant differentiation levels [(CC), (SF), (GV, TR), (AL, LCA, SLP), (V), (PCH), (ML), (SAB, TIN), (CAT), (TE, TA, RTZ)].

## Isolation by distance, effective sample size, and gene flow

In the Mantel tests for both the 16 (all) and 12 (surface only) locations, the results showed a non-significant correlation ($R^2 = 0.07$, $P = 0.908$; $R^2 = 0.067$, $P = 0.125$, respectively). A fourth Mantel test, using all surface locations except for CAT (11 locations), also recorded a significant correlation between genetic and geographic distances ($R^2 = 0.633$, $P = 0.0$; Fig. S1B).

The smallest values of the effective population sizes (Ne) were presented by caves locations (average values from 1,153.334 ML to 3,260 for TIN; Fig. 3). For surface locations, the lowest Ne values were recorded in northern localities (average values of 5,316.667 for CC and 10,816.667 for SF; Fig. 3). Remaining surface locations presented higher Ne values (average values from 16,677.78 for LCA to 22,450.00 for TE; Fig. 3).

In general, the historical migration rates (m) among locations and populations show low values (below $m = 0.005$), except for the highest genetic flow values recorded among distinct basins. Of particular note is the TR location from the Soto la Marina basin that presented a symmetric flow with the CC population, and the same TR that presented an asymmetric flow toward the cave population PCH. All three of these populations are assigned to *A. mexicanus* (Fig. 4). Intermediate gene flow values also were recorded among distinct basins (Fig. 4). It should be noted that in the lowest gene flow there were no differences within basins, between basins, or between caves (Fig. 4).

Considerable recent gene flow using BAYESASS 3.04 was detected only unidirectionally among the following locations within the same cave or basin system that formed a single

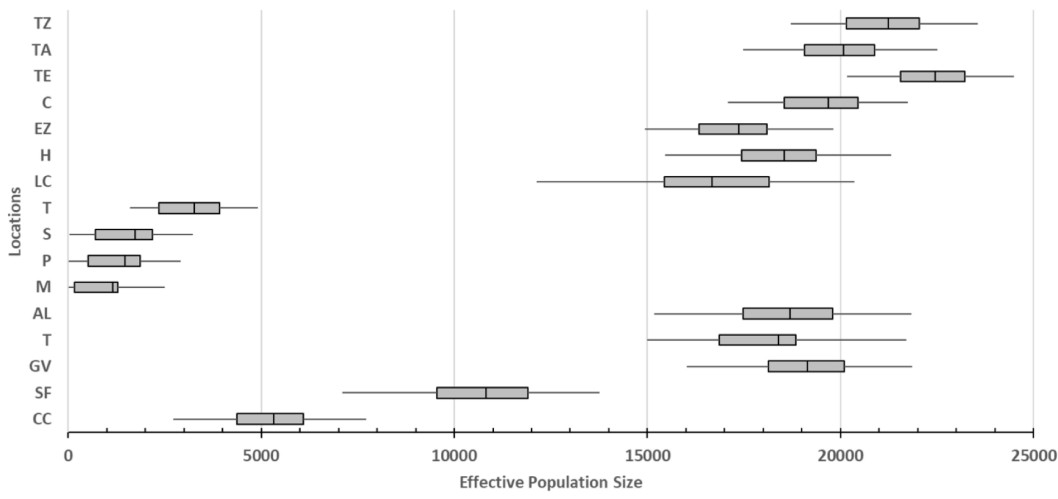

**Figure 3** **Effective population size of 16 locations sampled based on Bayesian inferences and obtained by the product with the mutation rate = 5 × 10⁻⁴ (*Fumey et al., 2018*).** For definition of the acronyms see Table 1. The central box of the plots represents the values from the lower to upper quartile (25 to 75 percentile). The middle line represents the median posterior values over all loci. The horizontal line extends from the 2.5% percentile to the 97.5 percentile.

population: for cave locations, from TIN to SAB, and for surface locations, from GV to TR in the Soto la Marina River basin, from SLP and LCA to AL in the Pánuco River basin, and from TA to TE in the Grijalva-Usumacinta River basin (Table 4).

## Characterizing population genetic variability

The ten populations varied in terms of the polymorphism of the microsatellite loci. The number of alleles per locus ($N_a$) ranged from 1 to 23 (average = 8.30), number of effective alleles ($N_{ae}$) ranged from 1 to 11.8 (average = 4.0), allelic richness ($A_R$) ranged from 1 to 13.9 (average = 5.8), observed heterozygosity (Ho) presented values from 0 to 0.923 (average = 0.485), expected heterozygosity (uHe) ranged from 0 to 0.921 (average = 0.556), and the fixation index ($F_{IS}$) ranged from −0.292 to 0.665 (average = 0.145). As expected, the cave populations (ML, PCH, SAB/TIN) presented the lowest values of genetic variability and the highest values of the fixation index, while the surface populations showed higher variability and lower values of the fixation index (Fig. S2 and Table S3).

## DISCUSSION

The *A. mexicanus* from the Gulf of Mexico slope comprises what is probably one of the most studied groups of cavefish (*Gross, 2012*; *Ornelas-García & Pedraza-Lara, 2016*). However, the relationship between these cavefish and their surface counterparts is still poorly understood, especially among populations located in the northernmost ranges. These populations have been sub-sampled only for use in phylogeographic studies using genetic sequences of nuclear (*Ornelas-García, Dominguez-Dominguez & Doadrio, 2008*) and mitochondrial (*Strecker, Faundez & Wilkens, 2004*; *Ornelas-García, Dominguez-Dominguez & Doadrio, 2008*) loci, as well as SNPs (*Coghill et al., 2014*). Though recent studies based on

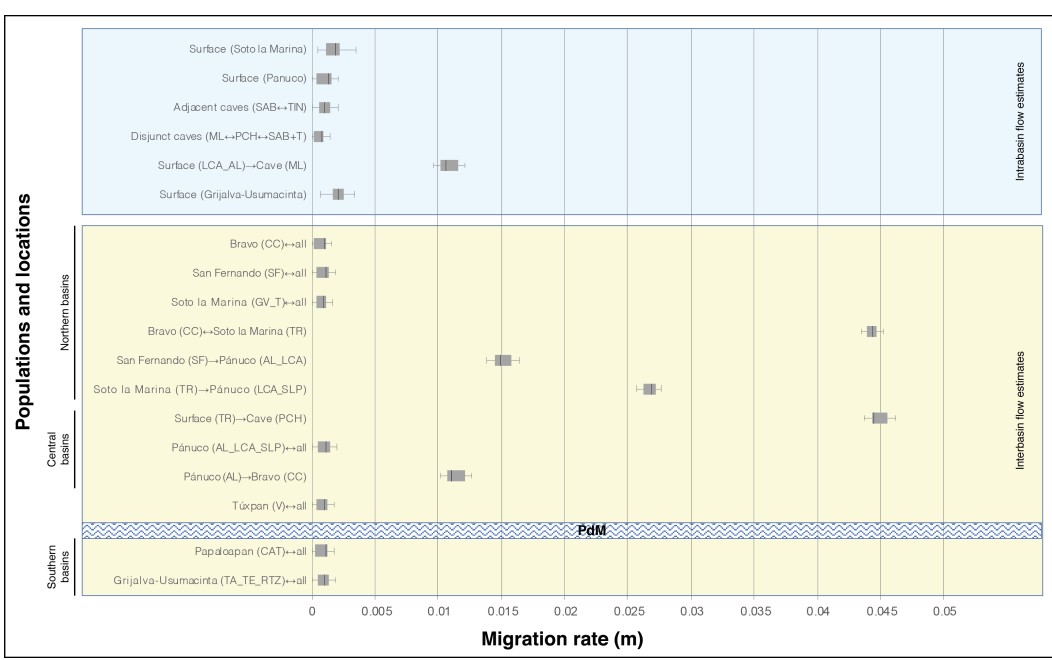

**Figure 4** **Migration rate (m) based on Bayesian inference among distinct locations of *A. mexicanus* and obtained by the product with the mutation rate = 5 × 10⁻⁴ (*Fumey et al., 2018*).** For definition of the acronyms see Table 1. The central box of the plots represents the values from the lower to upper quartile (25 to 75 percentile). The middle line represents the median posterior values over all loci. The horizontal line extends from the 2.5% percentile to the 97.5 percentile. Horizontal texturized bar represents to biogeographic barrier ''Punta del Morro'' (PdM).

**Table 4** **Summary of BAYESASS analysis.**

| Source site | Target site | m | Lower 95% CI | Upper 95% CI |
|---|---|---|---|---|
| GV | TRO | 0.2057 | 0.1664 | 0.2450 |
| TIN | SAB | 0.1552 | 0.0977 | 0.2127 |
| LCA | AL | 0.2083 | 0.1685 | 0.2481 |
| SLP | AL | 0.1807 | 0.1360 | 0.2254 |
| RTZ | TE | 0.2453 | 0.2737 | 0.2169 |
| TA | TE | 0.2178 | 0.2532 | 0.1824 |

**Notes.**
Considerably higher mean values of migration rates (m) with their 95% confidence intervals are shown. m, migration rate, CI, confidence interval. For definition of the acronyms, see Table 1.

powerful genomic approaches have provided important insights regarding hybridization between fish lineages (between cave and surface, and among caves), the geographic extension and reticulate evolution of this study model continues to be a challenge in terms of explaining the evolution of the complex *Astyanax* cave-surface populations or lineages (*Herman et al., 2018*). This confirms the importance of the different efforts and approaches utilized in order to fully understand this model. The present study's extensive sampling throughout the geographic distribution range of the *A. mexicanus* of the Gulf of Mexico

slope proved an invaluable advantage. In addition to defining the number and genetic variation of populations within this region, it has revealed the contemporary and historical dynamic of the genetic exchange that determined the populations.

This study revealed 10 surface and cave genetic populations or lineages (Fig. 1). According to the $F_{ST}$ values, five levels of genetic differentiation (Table 2) were distinguished along the Gulf of Mexico slope, from the Grijalva-Usumacinta system in southern Mexico, to the Bravo River basin in northern Mexico (Tables 2; 3; Figs. 1; 2). These results clearly show that the pattern of population structure obtained is not consistent with the assignment to the three taxa mentioned above (Table 1), and is consistent with the previously stated taxonomic uncertainties (*Hausdorf, Wilkens & Strecker, 2011*; *Herman et al., 2018*).

At the highest differentiation level ($F_{ST} = 0.77$ to 0.85), the three cave populations differentiated considerably (Figs. 2A and 2B), with the highest probability of assignment based on the STRUCTURE analysis, ML: 99%, PCH: 99% and SAB+TIN: 97%, and no evidence of admixture between them (Fig. 1). This suggests almost complete isolation among cave populations from different geographic areas, such as the case of ML (from Sierra de Guatemala), and PCH, and SAB+TIN (from Sierra de El Abra), as well as among cave populations within the same geographic area (Sierra de El Abra). The pattern of similarity between the SAB and TIN caves could be explained by the physical connectivity between these two, since the SAB+TIN caves form the "*Sistema de los Sabinos*", a cluster comprising the Sabinos, Arroyo, and Tinajas caves, which are connected by sumps (*Elliott, 2016*). Other studies found a similar differentiation pattern between cave populations; mainly between the regions of El Abra and Sierra de Guatemala (*Strecker, Hausdorf & Wilkens, 2012*; *Bradic et al., 2012*), and between caves that are geographically distant from each other within the El Abra region (*Bradic et al., 2012*). In particular, the structure and differentiation between the PCH and SAB+TIN cave populations are consistent with mitochondrial lineages (*sensu Ornelas-García, Dominguez-Dominguez & Doadrio, 2008*) and SNPs markers (*Coghill et al., 2014*) that suggest two independent invasions within the El Abra region (see Confirming multiple independent cave invasions section).

The second level of genetic differentiation ($F_{ST} = 0.30–0.67$) also revealed a marked separation between cave and surface populations (Table 2), indicating a strong isolation among the surface and cave populations, although there were two cases of historical gene flow from surface to caves (see below). It is important to point out that the present study did not include any caves with an admixture of both surface and cave morphs, as is the case of the Caballo Moro, Chica, Micos, and Yerbaniz caves where a high frequency of surface fish is observed (*Strecker, Hausdorf & Wilkens, 2012*; *Bradic et al., 2012*).

The third level of genetic differentiation ($F_{ST} = 0.11–0.27$) was observed among the following seven genetic clusters of surface populations, revealing a hydrographic divergence pattern: CC (Bravo basin), SF (San Fernando basin), GV+TR (Soto la Marina basin), V (Tuxpan basin), AL+SLP+LCA (Pánuco basin), CAT (Papaloapan basin), and TE+TA+RTZ (Grijalva-Usumacinta basin) (Table 2; Fig. 1). Likewise, this called attention to the fact that the genetic divergence pattern lacks geographic congruence, which was supported by the Mantel test ($R^2 = 0.067$, $p = 0.125$) in which all 12 surface locations were analyzed. Thus, the pattern of differentiation among surface populations is structured

by the hydrographic basins along the Atlantic slope of Mexico. Based on SNPs nuclear loci and practically the same location sampling as used in the present study, *Coghill et al. (2014)* recovered most of these genetic surface populations as independent lineages of well-structured populations. However, we want to highlight the case of the CAT population that was previously considered to be a different genus (*Bramocharax*). Recent findings confirmed that this population belongs to the genus *Astyanax* (*Ornelas-García, Bastir & Doadrio, 2014*; *Garita-Alvarado, Barluenga & Ornelas-García, 2018*; *Ornelas-García et al., in press*). Although CAT recorded low gene flow in relation to other locations, the high diversity and the heterogeneous structure pattern (Fig. 1) suggest an ancient gene exchange that preceded the isolation of Catemaco Lake. This could explain the conspicuous morphological differences related to trophic specialization (body shape and the trophic traits of the head, *Garita-Alvarado, Barluenga & Ornelas-García, 2018*; *Ornelas-García et al., in press*) within this population. This is in accordance with the lack of genetic structure between the two differentiated morphs within Catemaco Lake, whose origin had been associated with a process of ecological speciation (*Ornelas-García et al., in press*). The heterogeneous structure pattern in addition to low gene flow, both found herein, are consistent with the relevant role of ancient hybridization in the repeated evolution of traits (*sensu Herman et al., 2018*).

The fourth level of genetic differentiation ($F_{ST} = 0.05$–$0.10$) involved the AL, SLP, and LCA locations within the Pánuco basin, and locations from the Grijalva-Usumacinta population, in addition to the CAT population in relation to the Grijalva-Usumacinta population (Table 2; Figs. 1; Figs. 2C and 2D). These populations, recognized as two mitochondrially well differentiated lineages that correspond to the *A. mexicanus* and *A. aeneus* lineages (*Ornelas-García, Dominguez-Dominguez & Doadrio, 2008*; *Hausdorf, Wilkens & Strecker, 2011*), show high levels of genetic mixing and reciprocal introgression (Fig. 1), highlighting the taxonomic uncertainties between these species. Using different microsatellite loci and locations from the Pánuco and Papaloapan River basins, *Hausdorf, Wilkens & Strecker (2011)* found a similar introgression pattern, confirming genetic mixing between populations of these two independent basins, despite the physical obstacle presented by the TMVB. The fifth level of genetic differentiation, which presented the lowest and non-significant values ($F_{ST} = 0.01$–$0.02$), occurred only among surface populations within the same river basin, particularly in the northern Soto la Marina and southern Grijalva-Usumacinta basins (Table 2; Figs. 1; Figs. 2C and 2D).

## Confirming multiple independent cave invasions

Until a few years ago, the widely accepted evolutionary scenario regarding the origin of *A. mexicanus* cavefish was that some of these populations were ancient relict populations (i.e., hundreds of thousands of years to millions of years) whose surface relatives were extinct, while other cave populations were considered more recent, and therefore closer to current surface fish populations. This scenario involved two types of cave populations derived from an ancient invasion and a recent invasion (revision in *Gross, 2012*). However, recently *Fumey et al. (2018)* found that the origin of the blind cave fish is much more recent (<20,000 years), thus refuting the hypothesis of the two ancient invasions. In another study

based on independent approach, the findings also refuted the origin theory of two ancient invasions, although they supported at least four independent origins for the cavefish populations (*Coghill et al., 2014*). *Bradic et al. (2012)*, while recognizing the origin of the cave populations through the two ancient invasions, also found four independent invasion events within the Sierra del Abra (Pachón, Chica cave, Micos, and the six adjacent central caves [Yerbaniz, Japones, Arroyo, Tinajas, Curva, and Toro]) which correspond to the oldest invasion wave, plus a fifth invasion within the Sierra de Guatemala. The isolation and differentiation pattern of the cave populations included in the present study, ML, PCH, and SAB+TIN caves (Figs. 1; 2A and 2B; Table 2), indicate three independent origins, thus, corroborating those findings pointing to multiple independent subterranean invasions. Greater efforts in field research and modern genomic techniques will surely be required to accurately determine the invasion times of this highly dispersive species into caves.

The case of the SAB+TIN cave fish, while they were not separated by the STRUCTURE and DAPC analyses (Figs. 1; Figs. 2A and 2B), both presented considerable and significant differentiation coefficients ($F_{ST} = 0.27$, Table 2). Despite the physical connectivity between these caves, they are not adjacent and are in fact located at different elevations, with the Sabinos cave located at a higher elevation (239 masl) than the Tinajas cave (166 masl) (*Elliott, 2016*). Such inconsistency in the level of structure (Figs. 1; Figs. 2A and 2B) could be the result of two possible scenarios regarding how the SAB+TIN complex was formed. The first (*e1*) considers that a cave population within "*Sistema de los Sabinos*" experienced a partial fragmentation, implying the occurrence of an isolation stage with low connectivity as indicated by the historical gene flow (Fig. 4). The second (*e2*) scenario consists of independent isolation events in each of these caves derived from a common surface ancestor, which would imply two independent cave invasions in "*Sistema de los Sabinos*", with subsequent secondary contact. These scenarios could both reflect the very dynamic karst system in northeastern Mexico, associated with the formation of new caves, and/or incidents of separation or fusion of these systems (*Strecker, Hausdorf & Wilkens, 2012*) (see below for more discussion).

## Dispersal pattern of *Astyanax mexicanus* throughout the Gulf of Mexico slope

In concordance with previous studies (*Bradic et al., 2012*; *Fumey et al., 2018*), findings obtained herein showed very low gene flow in both intra-basin locations and populations, and inter-basin populations. However, there were six particular cases that recorded the highest migration rates, which will be discussed next (Fig. 4). Recent findings based on analysis of the whole genome, support the occurrence of considerable historical and contemporary gene flow between cave and surface populations (*Herman et al., 2018*).

### Migration among surface populations

Although most of the historical migratory events between surface populations, including intra and inter-basin gene flow, recorded lower gene flow (below m = 0.005), there were four cases of moderate to high gene flow (Fig. 4). The highest and symmetrical rates were found in northern locations between populations in disjunct basins, the CC population on the Bravo basin and the TR river, a tributary of the Soto la Marina River basin (Fig.

4). Considerable differences in migration rates between these two populations in relation to those observed among other remaining surface populations, suggest the existence of an ancestral connection between the Soto la Marina River and the Cuatro Ciénegas valley. Other species of primary freshwater fish distributed among the Soto la Marina, Cuatro Ciénegas valley, and other independent basins (*Minckley, 1984*; *García-de León et al., 2005*) support such an ancestral connection, e.g., *Ictalurus lupus* (Girard 1858), *Herichthys cyanoguttatus* (Baird & Girard 1854), and *Micropterus salmoides* (Lacepède 1802). There are two likely connections: (1) a hypothetical ancestral corridor through the intermediate headwaters of the San Fernando River Basin, and (2) an ancestral coastal system like the current Laguna Madre, an extensive coastal lagoon into which the Bravo, San Fernando, and Soto la Marina rivers flow (*De la Lanza-Espino, Ortiz-Pérez & Carbajal-Pérez, 2013*). The tolerance to salinity observed in some species of *Astyanax* has allowed them to inhabit coastal lagoons  (*Wilkens, 1982*; *Avilés-Torrez, Schmitter-Soto & Barrientos-Medina, 2001*; *Zubiria-Rengifo et al., 2009*). This indicates that an ancestral coastal system may have functioned as a corridor that connected only the Bravo and Soto la Marina rivers, thus allowing considerable historical gene flow.

The other three gene flow estimates were moderate and asymmetric, and all three included the Pánuco River basin. In two cases the direction of flow was from northern populations of the San Fernando (SF population) and Soto la Marina (TR population) basins towards southern populations (AL, LC and SLP) in the Pánuco basin (Fig. 4). The other unidirectional migration event was in the opposite direction, from the population of the Pánuco basin (AL) to the population of the Bravo basin (CC). The existence of approximately 15 coastal lagoons and the more than 40 estuaries that currently lie between the Bravo and the Pánuco river basins suggests that the historical dynamic of these coastal systems, which involves large changes in their geomorphology, caused extreme fluctuations in environmental conditions (*De la Lanza-Espino, Ortiz-Pérez & Carbajal-Pérez, 2013*), and likely had an influence on the gene flow between the populations of the northern and southern basins. In an apparently remote scenario of connection, *Herman et al. (2018)* found historical gene flow between two geographically distant populations, the Rascón surface population (from a Pánuco basin) and one population of *A. aeneus s. stricto* from Guerrero, Mexico in the Pacific slope. Further, in the same study genetic exchange was found among the *A. aeneus* population and the Choy surface population (also from the Pánuco basin) and ML cave population. Rather than regarding these two gene flow events as evidence of direct connection between populations, the authors interpreted them as signals of ancient hybridization in the genome. Based on the above, an alternative explanation to the scenarios of the connections discussed above, is that the historical gene flow signal observed in the present study, indicates a past genetic exchange in a common ancestral area where the lineages that preceded the present populations met. Finally, the low estimates of historical gene flow recorded between populations located north and south of "Punta del Morro" confirm that this volcanic mountain is a biogeographic barrier to freshwater fish (*Contreras, Obregón & Lozano, 1996*; *Hulsey et al., 2004*).

The significant and unidirectional contemporary gene flow between surface populations observed only between locations in the same basin (within the Soto la Marina, Pánuco, and

Grijalva-Usumacinta basins), suggests a continuous corridor between source and target locations.

### Migration among cave populations

The historical gene flow found between the adjacent SAB and TIN caves was symmetrical and low, but slightly higher than gene flow among disjunct caves (Fig. 4). The low historical gene flow is related to the moderate and significant $F_{ST}$ value (0.27). Estimates of both the differentiation index and gene flow gave results similar to those recorded among surface populations from different river basins (Table 2; Fig. 4). Thus, although the genetic structure results (Fig. 1; Figs. 2A and 2B) indicate that the SAB and TIN caves contain a single population, the level of differentiation between these two caves is striking. The Tinajas and Sabinos caves have temporally independent origins; the former is the oldest and most distant cave of the "*Sistema de los Sabinos*" (*Elliott, 2016*). Based on the above, scenario *e2* is the most plausible, implying another independent cave invasion within the El Abra region, with subsequent secondary contact through the fusion of caves. With respect to contemporary gene flow, rather than being associated with the rise of a barrier between Sabinos and Tinajas, the unidirectionality from TIN to SAB (Table 4) could be associated with a demographic factor, i.e., the Tinajas cave has two large lake passages (*Elliott, 2016*), which potentially house a larger population that directly affects the proportion of migration rate. Regardless of flow direction, it should be noted that since these two cave populations have become connected, they have experienced continuous gene flow over time, increasing towards the present day.

### Migration from surface to cave populations

In this case, significant estimates were only obtained for historical gene flow. There were two cases of dispersal from surface to cave populations (Fig. 4). The first was a moderate intra-basin gene flow from the LC and AL corresponding to a surface population to the ML cave population, both located within the Pánuco River (Fig. 4). The second was a higher inter-basin gene flow in the TR location, from the Soto la Marina basin to the PCH cave population in the Pánuco basin (Fig. 4). The first case, which implicates intra-basin gene flow, is highly consistent with findings obtained in *Bradic et al. (2012)*, which recorded higher gene flow from surface populations found very close to the cave populations. Furthermore, contemporary and historical gene flow between surface and cave populations recorded recently is associated with the intermediate phenotypes observed in several caves that resulted from the surface introgression into cave populations. This surface introgression is promoted by flooding in caves during the rainy season and is considered to play an important role in a repeated evolutionary adaption in cavefish (*Herman et al., 2018*). In the second case of gene flow between adjacent basins (Soto la Marina and Pánuco), the high estimate (Fig. 4) could indicate an ancient admixture within a proximate area. An alternative explanation could be the presence of an ancient TR surface migrant within the Bravo and Pánuco basins, suggesting that the Soto la Marina basin functioned as a source population that expanded to the northwest and south, as shown by our estimates of gene flow (Fig. 4).

## CONCLUSIONS

The results of this study, which used a particular set of 10 microsatellite loci for the genus *Astyanax*, confirmed previous findings in *A. mexicanus* from the Gulf of Mexico slope, obtained with a different set of microsatellite loci as well as other molecular markers and samples, and provided new findings.

Ten discrete populations or lineages consisting of seven surface and three cave populations with different levels of differentiation and structure were detected, with the highest levels presented when cave populations were compared to each other, or with surface populations. This pattern confirms that the cave populations, particularly those included in this study such as the Molino cave from Sierra de Guatemala and the Pachón, Sabinos, and Tinajas caves from the El Abra region, correspond to relicts of independent invasions. The findings obtained in this study revealed two independent invasions in the "*Sistema de los Sabinos*" cluster of caves.

In addition, the present study confirmed a substantial genetic exchange between distinct populations, including surface and cave populations, mainly between hydrographic systems located north of the volcanic mountain range "Punta del Morro". The inter-basin gene flow, at least the flow to the north of the "Punta del Morro", indicates the occurrence of bidirectional invasions, confirming the strength of this physiographic component as a biogeographic barrier. Consistently, the contemporary gene flow only occurred between populations within the same basin.

Finally, the heterogeneous structure pattern, in addition to the low gene flow recorded in the Catemaco Lake population, suggested an ancient hybridization that could have given rise to the two divergent lacustrine morphs. Likewise, the history of reticulate evolution within *A. mexicanus* by hybridization is confirmed.

## ACKNOWLEDGEMENTS

This study was developed with the collaboration of several national and international colleagues. We thank Dean Hendrickson, Hector Espinosa and Gil Rosenthal for providing samples. We also thank Rafael Hernández Guzmán and Veronica Mendoza Portillo for their invaluable help in the development of the distribution map and the performance of the Migrate analyses, respectively.

### Funding

Funds were received from FJGdL and CIBNOR's Genetics for Conservation Laboratory) and from the Posgrado de la Facultad de Biología, UMSNH.

### Grant Disclosures

The following grant information was disclosed by the authors:
CIBNOR's Genetics for Conservation Laboratory and Posgrado de la Facultad de Biologíá, UMSNH.

## Competing Interests

The authors declare there are no competing interests.

## Author Contributions

- Rodolfo Pérez-Rodríguez wrote the first draft and reviewed subsequent draft versions of the article, prepared figures and/or tables, coordinated project funding, and approved the final draft.
- Sarai Esquivel-Bobadilla performed the laboratory experiments, prepared figures and/or tables, sampling and data collection, and approved the final draft.
- Adonaji Madeleine Orozco-Ruíz and José Luis Olivas-Hernández analyzed the data, prepared figures and/or tables, and approved the final draft.
- Francisco Javier García-De León conceived and designed the experiments, coordinated project funding, authored or reviewed drafts of the paper, sampling, and approved the final draft.

## Field Study Permissions

The following information was supplied relating to field study approvals (i.e., approving body and any reference numbers):

Collection permits provided by the Mexican government (DGOPA.05003.181010-5003; DGOPA.00570.288108-0291; DAPA/2/130409/0961, 230401-613-03).

## Data Availability

Raw data, the location of each specimen, and the specimen identification/tissue voucher ID, are available in the Supplementary Files.

## Supplemental Information

Supplemental information for this article can be found online at http://dx.doi.org/10.7717/peerj.10784#supplemental-information.

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
