# Peer review of "Genetic structure and historical and contemporary gene flow of Astyanaxmexicanus in the Gulf of Mexico slope: a microsatellite-based analysis"

_PeerJ, doi:10.7717/peerj.10784_

## Round 0.1 · original submission · Major Revisions

I have received two evaluation reports on the submitted manuscript. Despite the effort made in data collection and laboratory, the reviewers mention drawbacks and limitations, which fundamentally question the data presented and, hence, respective methodology/discussion. They provided very constructive comments on how the manuscript can be improved. Furthermore, I included some comments that should be considered. I hope that you will find all advice helpful when revising the manuscript.

(1) The Introduction should be better reframed, and the authors should include some clear hypotheses, which need to follow from your Introduction of the rationale. For instance, more information about the influence of rivers as putative barriers modulating gene flow between sampling locations are needed, and a more straightforward hypothesis (e.g., Drift paradox; Ritland K. 1989. Canadian Journal of Botany 67: 2017–2024) should be included.

(2) The use of the word ‘population’ throughout the manuscript is not appropriate. A typical biological definition of ‘population’ is a group of interbreeding individuals that share time and space (Hedrick PW (2000) Genetics of Populations, 2nd edn. Jones and Bartlett, Sudbury, Massachusetts.), and most definitions involve some type if interbreeding. One of the goals of population genetics is to identify what is a population and how many are present. Many statistical tests were developed to identify populations. We need to be precise with our terminology and you need to demonstrate that the ‘units of sampling’ are sufficiently distinct to be considered as separate populations. It would be better if ‘sampling location’ or ‘site’ is used.

(3) As the sample sizes are different between sampling locations, authors need to be more careful with the analyses. Are the genetic diversity parameters, obtained from different sample sizes, comparable among sampling locations? For instance, the authors used “richness allelic” but not was mentioned regarding the rarefaction procedure; please clarify it. Besides, as you have small sample sizes (i.e., n<50), the unbiased gene diversity should be used instead of ‘expected heterozygosity’.

(4) As also pointed out by the reviewer #2, more details about the methods should be informed. Additional points: (i) AMOVA models should be better specified and not just presented in the Supplementary material. (ii) Regarding the historical migration rates, the authors should inform the mutation rate used to calculate ‘m’. In addition, double-check the ‘m’ values presented in Table 3; migration rates for some ‘sampling locations’ are very high and unusual. (iii) As were identified the presence of null alleles, authors should use appropriate programs wich taking it into account. For example, the ‘ENA’ algorithm as implemented in FreeNA (Chapuis & Estoup 2007) can be used to estimate population pairwise FST.

(5) The authors should improve the quality of the tables and figures presented.

Reviewer 1 ·

Basic reporting

In general, the manuscript is well written and clear, with good coverage of literature information. The main results are discussed in order to answer the questions proposed by the authors.

Experimental design

The questions proposed by the authors are simple and do not provide any expectations a priori or hypotheses to be tested. I suggest that an improvement should be made in relation to the main question, the expected results, and the hypotheses to be investigated. Nevertheless, the methods applied are quite appropriate to investigate patterns of population structure and gene flow (both contemporary and historical). Despite the low number of genetic markers used (10 microsatellites), the authors compensate with an extensive number of samples.

Validity of the findings

The authors investigated the genetic structure, as well as the current and historic gene flow of the characid fish genus Astyanax spp. by examining the number of genetically differentiated populations existing in the different basins and caves of the Gulf of Mexico slope, as well as the degree of connectivity between such populations. However, since this is a population study with a group that probably has different lineages/species involved in complex processes of isolation, expansion, and hybridization, the discussion is confused in terms of which of these `species' may be involved in the evolutionary events suggested by the authors (more general comments below). In addition, based on contemporary and historical gene flow information, the authors discuss the possibility of different dispersion patterns between surface and caves. I suggest special attention with this interpretation, because historical isolation and the difference in physical connectivity between the two environments (surface and cave) can bias the gene flow pattern in the group, and not be necessarily linked to the dispersion ability of the organism in space. Therefore, to guarantee the best experience for the PeerJ readers, I recommend that these and other minor revisions should be made before considering the present manuscript for publication.

Additional comments

Major revisions:
1# Despite the phylogenetic controversies about the group, the authors investigate the genetic structure of the population in the genus Astyanax spp. I suggest that the authors adopt the most coherent phylogenetic hypothesis available in the literature and indicate a priori which possible lineages (i.e., species) may be involved in the study and in which locations they belong and which genetic clusters they are assigned. That is, for better clarification it would be helpful if these lineages/species are delimited in Figures 1, 2, 3. The lack of hierarchical delimitation of evolutionary units under investigation (i.e., population, lineages, species, etc.) may confuse the reader and hide important biological aspects such as where and which lineages are involved in hybridization and colonization processes.

2# In the section 'Dispersal pattern of Astyanax spp. throughout the Gulf of Mexico slope', Line 394: the authors consider the existence of different dispersion capacities of the species of the genus, concluding higher dispersion capacity for surface populations and low dispersion capacity for cave populations. However, although gene flow (contemporary and historical) is an indirect measure of an organism's ability to disperse (i.e., gametes dispersion) it can be biased by historical factors of landscape evolution and connectivity (e.g., long periods of physical isolation or temporary barriers). Surface basins may have maintained more connecting routes between populations over time than between populations in cave complexes, which indicates less connectivity of environments and not necessarily the low dispersion capacity of organisms. An example of this can be observed in the caves of the 'Sistema de los Sabinos', which presented historical and current gene flow that could be merely the result of the physical connection of these localities (line 431-433). In addition, isolation by distance often is a result of limited dispersion in continuous environments. Mantel's tests in this study indicated significant correlations (0.696 and 0.633, line 233-239) between genetic and geographical distance for some surface populations. In this case, I believe that determining the different dispersion capabilities across these taxa becomes inconclusive.

Minor revisions:
In the abstract, it is not clear which are the main objectives/issues of the research. Please make it explicit.

Figure 1 - The authors labeled the localities as 'surface and cave populations', which is confusing with the definition of 'population' used in the text to express the observed genetic differentiation. For better clarification, I suggest replacing the term 'population' in Figure 1 by ' sampled localities'.

Figures 2 and 3 - For a better reference of which genetic groups belong to the surface or cave, I suggest the addition of some indicator in the figure that facilitates the rapid distinction by the readers.

Reviewer 2 ·

Basic reporting

In my opinion the authors need to better describe the morphological and ecological variations within the Astyanax complex. Describing the most important habitats and morphological traits. The author describe this for the caves taxa, but little was said about surface populations.

I also think the authors could better detail the characteristics of the landscape where the study is applied providing a better view on the spatial separation of populations from the caves and the surface. Which ecological factors could influence the genetic variation among populations?

A large part of the discussion is related to the different invasions that caves populations have undergone over time. However, only a few lines in the introduction give the reader a good background on this point.

In the introduction it is also mentioned that several studies of population genetics have already been carried out with this group. Is there any hypothesis to be raised about the direction of gene flow and connection based on these studies?

Experimental design

The research question of the study is well defined, and the methods to address them are ideal. However, a more detailed description of the analysis procedures must be presented.

Below I list some specific observations:

Line 96: I did not understand why the following sentence is cited in the text: "Some of these sites with fewer individuals had previously been studied in Coghill et al. (2014)."

Line 125: DAPC: The authors need to mention that they performed 2 types of analyzes, one with the caves populations and the other without them.

Line 136: AMOVA. I understand the reason for analyzing different groups of populations in AMOVA, but I would like the authors to explain the reason for the specific choice of the models tested. They tested 4 models (a,b,c and D; Table S2), and they need to be explained in the methods.

Line 145: Since its maximum number of populations is 16, why not test k ranging from 1-16?

Line 159: MIGRATE: The authors should make it clear which inference they used. Also, the choice of different comparisons between populations must be detailed and justified here in the methods.

General point: I would combine the topics Statistical analyzes and population genetic diversity into one topic, perhaps called genetic diversity

Validity of the findings

The results and Discussion provide an important advance in the study of this group with so many interesting evolutionary characteristics. But some parts of the results also deserve to be better explored.

Below I list some specific observations:

Line 194: The authors could also mention the few comparisons that were not significant.

Line 209: Four or five clusters???

Line 215: AMOVA: I would like the authors to better explore the results of AMOVA,
which means that model D has the least within group variance and the largest between groups?

Line 231: These procedures of including or excluding populations is also not described in the methods

Line 234: The Trans-Mexican Volcanic Belt should be shown in Figure 1

Line 237: Why except Catemaco???

line 240: Again, these different procedures using MIGRATE-N is not explained in the methods.

In fact, this part of the results is very difficult to understand. I didn't quite understand how the comparisons in table 3 were chosen.

-Table 3: It is TR or TRO???

-Why PCH is not included in the caves populations group (Table 3)?

-What are the comparisons called surface & Cave locations? why not use all cave pop???

-How are the genetic clusters found in the present study related to the three possible lineages (A. mexicanus, A. aeneus and A. hubbsi) mentioned in the Introduction???

- A more in-depth discussion of the genetic structure of populations would be welcome

- All table and figures legends could be improved.

Figure 2: Why not just leave the plots present in parts A´ and B´?

---

## Round 0.2 · accepted · Accept

The authors have done a good job revising the manuscript, and I appreciate the efforts that you have made in responding to the concerns pointed out by reviewers.

Reviewer 1 ·

Basic reporting

no comment

Experimental design

no comment

Validity of the findings

no comment

Additional comments

The revised manuscript presented a significant improvement in relation to the previous version. The research objectives are clear and the introduction presents a concise context about the investigated issue. Figures and Tables were improved according to the reviewers' suggestions, which made the visual information clearer. In the discussion the authors corrected information that previously seemed ambiguous or out of context, being now more coherent with the hypotheses about the evolutionary processes that determined the observed genetic patterns. Therefore, I believe that after the extensive review the manuscript finds the minimum criteria to be published in PeerJ. This is an important contribution to understanding the geographic and ecological processes that drove the genetic structure and dispersion (with subsequent hybridization) of the three lineages that compose the Astyanax group in the Gulf of Mexico basin.

Reviewer 2 ·

Basic reporting

The authors took into account most of my previous comments .

Experimental design

The methodology seems adequate

Validity of the findings

Suggestions were incorporated

Additional comments

The authors took into account most of my previous comments in the Introduction, except the one about the landscape characteristics. I still have doubts about how these characteristics (geographic, ecological, etc.) would affect the connection between populations and I think that a brief description of this would be interesting in the Introduction, but I respect their opinion in thinking that this is not fundamental. The methodology now seems adequate, all my suggestions were incorporated, figures also were improved.

Discussion
Line 417: supported is repeated